# Multi-Omic Analysis Reveals Different Effects of Sulforaphane on the Microbiome and Metabolome in Old Compared to Young Mice

**DOI:** 10.3390/microorganisms8101500

**Published:** 2020-09-29

**Authors:** Se-Ran Jun, Amrita Cheema, Chhanda Bose, Marjan Boerma, Philip T. Palade, Eugenia Carvalho, Sanjay Awasthi, Sharda P. Singh

**Affiliations:** 1Department of Biomedical Informatics, University of Arkansas for Medical Sciences, Little Rock, AR 72205, USA; SJun@uams.edu; 2Departments of Oncology and Biochemistry, Molecular and Cellular Biology, University Medical Center, Washington, DC 20057, USA; Amrita.Cheema@georgetown.edu; 3Department of Internal Medicine, Division of Hematology & Oncology, Texas Tech University Health Sciences Center, Lubbock, TX 79430, USA; chhanda.bose@ttuhsc.edu (C.B.); Sanjay.awasthi@ttuhsc.edu (S.A.); 4Division of Radiation Health, University of Arkansas for Medical Sciences, Little Rock, AR 72205, USA; MBoerma@uams.edu; 5Department of Pharmacology and Toxicology, University of Arkansas for Medical Sciences, Little Rock, AR 72205, USA; ppalade@uams.edu; 6Center for Neuroscience and Cell Biology, University of Coimbra, 3004-531 Coimbra, Portugal; eugeniamlcarvalho@gmail.com; 7Department of Geriatrics, University of Arkansas for Medical Sciences, Little Rock, AR 72205, USA

**Keywords:** aging, sulforaphane, gut microbiome, metabolome, biomarkers

## Abstract

Dietary factors modulate interactions between the microbiome, metabolome, and immune system. Sulforaphane (SFN) exerts effects on aging, cancer prevention and reducing insulin resistance. This study investigated effects of SFN on the gut microbiome and metabolome in old mouse model compared with young mice. Young (6–8 weeks) and old (21–22 months) male C57BL/6J mice were provided regular rodent chow ± SFN for 2 months. We collected fecal samples before and after SFN administration and profiled the microbiome and metabolome. Multi-omics datasets were analyzed individually and integrated to investigate the relationship between SFN diet, the gut microbiome, and metabolome. The SFN diet restored the gut microbiome in old mice to mimic that in young mice, enriching bacteria known to be associated with an improved intestinal barrier function and the production of anti-inflammatory compounds. The tricarboxylic acid cycle decreased and amino acid metabolism-related pathways increased. Integration of multi-omic datasets revealed SFN diet-induced metabolite biomarkers in old mice associated principally with the genera, *Oscillospira*, *Ruminococcus*, and *Allobaculum*. Collectively, our results support a hypothesis that SFN diet exerts anti-aging effects in part by influencing the gut microbiome and metabolome. Modulating the gut microbiome by SFN may have the potential to promote healthier aging.

## 1. Introduction

The gut microbiome is an extremely diverse and complex ecosystem of bacteria, viruses, and fungi inhabiting the intestinal tract, which interact with each other and their host [1]. The human gut microbiome is unique to each individual, but is dominated by anaerobic bacteria belonging to two phyla, Firmicutes and Bacteroidetes [2]. It is becoming apparent that there are many complex interactions between the gut microbiome, immune system, and metabolism in health and disease [3,4]. Accumulating data support the contention that microbial metabolites play major roles in the regulation of the immune system [5]. Unsurprisingly, diet is a major modifiable factor shaping the gut microbiome structure and metabolic activity both in the short and long terms as reported in observational and intervention studies [6,7]. Given these associations, dietary factors may have significant therapeutic utility in modulating many interactions between the gut microbiome, metabolism, and immune system.

The human gut microbiome fluctuates over the individual’s life span, undergoing the most prominent deviations during infancy and old age [8]. Interestingly, our immune health shows similar fluctuation patterns like the gut microbiome and the most unstable state during infancy and old age [8]. Although the causal relationship between the gut microbiome and microbiome may serve as a target for anti-aging intervention. We hypothesize that diet can revert the gut microbiome to a younger composition. In this study, we tested this hypothesis with a sulforaphane (SFN)-containing diet in a mouse model of aging. We have previously published a detailed study indicating that SFN may prevent age-related loss of function in the heart and skeletal muscle [9]. Our studies suggest that in the old mice, SFN restored Nrf2 activity, mitochondrial function, cardiac function, exercise capacity, glucose tolerance and activation/differentiation of skeletal muscle satellite cells. Our results suggested that the restoration of Nrf2 activity and endogenous cytoprotective mechanisms by SFN may represent a safe and effective strategy to protect against muscle and heart dysfunction due to aging.

SFN is a compound in cruciferous vegetables that has been investigated for its anti-cancer, anti-aging, antioxidant, antimicrobial, anti-inflammatory, anti-diabetic, and neuroprotective properties [10,11]. Many studies have investigated SFN for its role in aging which have led to the link between SFN mechanisms of action and Nrf2 and NF-κB, key cellular transcription factors. SFN-induced activation of Nrf2 and inhibition of NF-κB result in the induction of redox-modulating genes and the inhibition of inflammation, respectively [12]. However, the effect of SFN on the gut microbiome has not yet been investigated in aging.

We hypothesize that administration of SFN reshapes the old gut microbiome into a younger composition. This may lead to an enrichment of beneficial bacteria that produce short chain fatty acids (SCFAs), known for their anti-inflammatory actions that may help improve aging-related pathologies. In this study, we focus on differences in impact of SFN on the gut microbiota in old mice compared to young mice. To disentangle the contribution of aging and non-aging related variables to the gut microbiome, we divided mice into four groups: an old control group receiving regular rodent chow, an old group receiving the same chow but supplemented with SFN, a young control group, and a young SFN diet group (Table 1). Fecal samples were collected immediately before (indicated in text and figures as 0 months) and 2 months after the start of SFN administration. We conducted multi-omic profiling to investigate the relationship between SFN diet, the gut microbiome, and metabolome. Our results identified global relationships and highlighted novel associations between SFN diet and microbiome structure and function, and metabolite biomarkers that may be modulated by gut bacteria.

## 2. Materials and Methods

### 2.1. Study Design and Sample Collection

This study conformed to the Guide for the Care and Use of Laboratory Animals of the National Institutes of Health and the work was performed in accordance with a protocol (IACUC# 646767-5, 17-11-2014) approved by the Central Arkansas Veterans Healthcare System Institutional Animal Care and Use Committee. Animals were housed in the Veterinary Medical Unit at the Central Arkansas Veterans Healthcare System in Little Rock, AL, USA. Young and old male C57BL/6 mice were obtained from aged rodent colonies of the National Institutes of Health (Bethesda, MD, USA). d,l-sulforaphane was purchased from Toronto Research Chemicals, 20 Martin Ross Avenue, North York, ON, Canada, M3J 2K8 (https://www.trc-canada.com/product-detail/?S699115) that does not contain any other natural isothiocyanates from plants. Young (6–8 weeks of age) and old (21–22 months of age) mice were fed TD 96,163 diet (essentially oxidant free, Teklad, Madison, WI, USA) (control groups) or TD 96,163 diet supplemented with sulforaphane (SFN) (442.5 mg per kg diet; treated groups), for 2 months. Immediately before and at the completion of the 2 months-administration of SFN and control diet administration, fecal pellets were collected from five mice per age and treatment. For this purpose, mice were individually placed in a Plexiglas box to obtain fresh fecal pellets. Fecal pellets were collected and stored at −80 °C until processing. Each pellet was divided into two parts under liquid nitrogen. One half was shipped to the University of California Los Angeles for 16S rRNA amplicon sequencing and the other half to Georgetown University for metabolomics as described in Casero et al. [13] and also described below.

### 2.2. Sample Preparation for Microbiome Analysis

Sample preparation and analysis was performed as described by us previously [13]. Briefly, genomic DNA was extracted using the PowerSoil DNA Isolation Kit (MO BIO Laboratories, Carlsbad, CA, USA) as per manufacturer’s instructions. 16S rRNA amplicon bacterial gene sequencing was performed using extracted genomic DNA as the template using a standard protocol. The PCR primers used in the study (F515/R806) targeted the V4 hypervariable region of the 16S rRNA gene, with the reverse primers including a 12-bp Golay barcode. Thermal cycling was performed in an MJ Research PTC-200 (Bio-Rad Inc., Hercules, CA, USA) with the following parameters: 94 °C for 5 min; 35 cycles of 94 °C for 20 s, 50 °C for 20 s, and 72 °C for 30 s; 72 °C for 5 min. PCR products were purified using the MinElute 96 UF PCR Purification Kit (Qiagen, Valencia, CA, USA). DNA sequencing was performed using an Illumina HiSeq 2500 (Illumina, Inc., San Diego, CA, USA), in paired-ended mode. Clusters were created using template concentrations of 4 pM and PhiX at 65 K/mm^2^. Sequencing primers targeted 101 base pair reads of the 5′ end of the amplicons and seven base pair barcode reads. Reads were filtered using the following parameters: minimum Q-score: 30; maximum number of consecutive low-quality base calls allowed before truncating: 3; and maximum number of N characters allowed: 0. All filtered V4 reads had a length of 150 bp.

### 2.3. Sample Preparation for Metabolomics Analysis

Fecal samples were processed by initially homogenizing in extraction solvent cocktail containing 50% methanol, 30% isopropanol, 10% water and 10% chloroform and internal standards to allow a broad-based extraction, followed by protein crash using 1:1 acetonitrile [14]. The samples were centrifuged and the supernatant was dried and resuspended in water containing 50% methanol for MS analysis. The samples were resolved using reverse phase chromatography using an Acquity UPLC (Waters Corporation, Milford, MA, USA) system online Xevo–G2-QTOF-MS (Waters Corporation USA, Milford, MA, USA) operating in positive and negative ion mode, the details of tune page parameters have been described before [15,16,17]. Several measures, including randomizing the sample queue, use of pooled quality controls and standards, were used to monitor data quality, retention time drifts and signal intensity.

### 2.4. Bioinformatic Processing with 16S rRNA Amplicon Sequencing Data

We first removed adapters (barcodes, forward primer: GTGYCAGCMGCCGCGGTAA; and reverse primer: GGACTACNVGGGTWTCTAAT) in demultiplexed paired-end sequencing data of 40 sequencing libraries from one batch using Cutadapt [18], which resulted in about 3% of reduction in sequencing depth. For microbiome bioinformatic processing, raw sequence data were then imported into QIIME2 [19]. Sequencing depth ranged from 67,885 to 283,004 with a mean of 199,196 and a median of 219,054. After inspecting the quality profiles, it was clear that the reverse read quality dropped off more severely than in the forward read. Accordingly, we trimmed the reverse reads at position 136 and removed chimeric reads by consensus method followed by denoising and merging with DADA2 [20] (via q2-dada2) which resulted in a table of 1746 amplicon sequence variants (ASVs) for 40 samples with the total frequency of 3,341,551. For taxonomic assignment, we used a classifier that has been pretrained on Greengenes database (v13_8) with 99% operational taxonomic units (OTUs) and primers used for amplification and the length of sequence reads. Taxonomy was assigned to ASVs using the q2-feature-classifier [21] classify-sklearn naïve Bayes taxonomy classifier against the Greengenes (v13_8) 99% OTUs reference sequences [22]. ASVs were filtered out through the following two steps: first, ASVs which belong to chloroplast at the class level or mitochondria at the family level were filtered out, and then only those ASVs with at least phylum-level assignment were only kept; second, low-abundance ASVs with frequency less than 0.0005% of reads in the total dataset were removed as recommended for Illumina amplicon data [23], which allowed us to perform differential abundance analysis with increased detection sensitivity. The two filtering steps resulted in a feature table of 1377 ASVs with a total frequency of 3,336,130, which was used for downstream analysis. All ASVs were aligned with MAFFT [24] (via q2-alignment) and a phylogeny was constructed using Fasttree2 [25] (via q2-phylogeny). For alpha and beta diversity analysis, alpha-diversity metrics (observed OTUs and Faith’s Phylogenetic Diversity [26]) and beta diversity metrics (weighted UniFrac [27], unweighted UniFrac [28], Jaccard distance, and Bray–Curtis dissimilarity) were used. Sample ordination based on principle coordinate analysis (PCoA) were estimated using q2-diversity after samples were rarefied (subsampled without replacement) based on rarefaction analysis by phylogenetic diversity index and species richness to the number of sequences of the sample with the least number of sequences (24,788). For alpha diversity significance test, the Kruskal–Wallis test was used to evaluate the effect of the experimental factors on the relative abundance at ASV level. Similarly, significant differences in beta diversity between groups were tested using permutational multivariate analysis of variance (PERMANOVA) with 999 Monte Carlo permutations. We generated an OTU table by clustering all ASVs into OTUs using Vsearch [29] (via q2-Vsearch) with a similarity threshold of 97% against the Greengenes (v13_8). The number of OTUs detected in 40 samples was 1047, and the mean frequency was 70,387. After removing rare OTUs with frequency less than 0.0005% of reads in the total dataset and then rarefying the OTU table at a depth of 21,221 sequences per sample based on rarefaction analysis by phylogenetic diversity index and species richness, the resulting OTU table provides a highly replicated, deeply sequenced dataset with 928 OTUs and mean frequency of 70,310. For functional analysis of the microbiome, we predicted functional profiles from the rarefied OTU table using PICRUSt2 v.2.1.4-b software [30]. With the maximum NSTI cut-off of 2, PICRUSt2 predicts abundance of Kyoto Encyclopedia of Genes and Genomes (KEGG) Orthology (KO) gene family and Enzyme Commission (EC) number based on the KEGG database [31]. Afterwards, we inferred the abundances of metabolic pathways based on MetaCyc database [32] from the predicted abundances of EC numbers. The linear discriminant analysis (LDA) effect size (LEfSe) method [33] was used to identify statistically significant differences between groups of experimental design in taxonomic and metabolic pathway features. For taxonomic feature, the ASV table was collapsed at the species level. LEfSe algorithm uses Kruskal–Wallis rank sum test to detect features with significantly different abundances between the two groups, followed by LDA to estimate the effect size of each feature. A strength of the LEfSe method compared with standard statistical approaches is that in addition to providing *p*-values, it estimates the magnitude of the association between each feature and the grouping categories. A significance alpha level of 0.05 for the Kruskal–Wallis test and an effect size (LDA score) threshold of 3 were used for all biomarkers. The cladogram from the LEfSe method indicates the phylogenetic distribution representing differentially abundant taxonomic groups. The size of each node represents its relative abundance.

### 2.5. Computation Frameworks to Integrate Microbiome and Metabolome for the Identification of Potential Mechanistic Links

We explored two different approaches to integrate microbiome and metabolome for the identification of potential mechanistic links: MIMOSA [34] using the updated software (https://borenstein-lab.github.io/MIMOSA2shiny/) and MelonnPan [35]. MIMOSA integrates metabolic potential from bacterial genomes and metabolome into a unified analysis. PICRUSt2-predicted metagenome from a reference-based OTU table, rarefied (described above), was used for the MIMOSA analysis. MIMOSA first performs metabolic network modelling using the Predicted Relative Metabolic Turnover framework [36] derived from KEGG enzymatic reactions [31]. For each metabolite in each sample, community-wide metabolic potential (CMP) scores were computed as the matrix multiplication of a stoichiometric enzyme reaction matrix (M) and PICRUSt2-predicted metagenome matrix represented by KO-relative abundances (G) so that CMP scores represent the relative capacity of the community in a given sample to generate or deplete each metabolite. The integration with metabolomics data was performed by comparing CMP scores to actual LC-MS normalized metabolite abundances, by matching metabolite putative ids with KEGG compound ids. Due to this algorithmic step, only putative metabolites which are annotated by KEGG compound id, can be examined for integration of microbiome and metabolome by MIMOSA. Finally, MIMOSA identifies contributions of taxa based on the amount of variation in a metabolite explained by CMP scores analyzing the regression model fit. For metabolites that are significantly associated with predicted metabolic potential with a regression model *p*-value less than 0.1, the taxa with the largest contributions are hypothesized to be the main drivers of change of metabolite pattern across samples. However, it is difficult to apply or validate MIMOSA results in a data-driven manner because of its algorithmic limitations of applicability to metabolites not annotated with KEGG compound. In addition, MIMOSA depends on accurate characterization and annotation of species- and even strain-specific metabolites so that it cannot scale well to complex communities with partially referenced taxa or metabolites. We applied another computational framework named MelonnPan which does not rely on a limited number of well-characterized taxa, enzymes, metabolites, and functional annotation unlike MIMOSA. MelonnPan uses elastic net regularization [37] to identify which OTUs are predictive for a given metabolite based on only their relative abundance profiles. For each metabolite, MelonnPan fits the elastic net model and optimizes the tuning parameters (i.e., both the elastic net mixing parameter and sparsity parameters) based on cross-validation. MelonnPan evaluates the predictability of each metabolite based on Spearman correlation coefficient (r) between the experimentally measured and predicted metabolite concentrations across samples. Metabolites with r > 0.3 are defined as well predicted. For significance testing on well-predicted metabolites, MelonnPan repeatedly shuffles the sample labels in both a metabolite table and an OTU table, and applies the MelonnPan to the randomized data by shuffling to link OTUs to metabolites and finally compares well-predicted metabolites between the original data and the randomized data. For multi-omics integration, we focused on statistically significant metabolites between OC2 and **OS2** since we are interested in bacterial populations that might be responsible for changes in metabolite pattern due to SFN diet in old mice.

## 3. Results

### 3.1. Data Annotation and Overview of Samples

Both aging and senescence are widely modelled in mice because of their physiological similarities to humans [38]. Male C57BL/6J mice at 6–8 weeks of age (corresponding to 17–20 years in humans) and mice at 21–22 months of age (60–65 years old in human age) were included in the study. From both cohorts, fecal samples were collected immediately before (0 months) and 2 months after the start of SFN-containing diet or control diet (Table 1) and stored frozen until microbiome and metabolome analysis. The fecal microbiome was analyzed by amplicon sequencing of the V4 region of the 16S rRNA gene. We obtained a total of 3.3 million high-quality amplicon sequence variants (ASVs) from 40 samples whose sequencing depth ranged from 24,788 to 130,121 with a mean frequency of 83,539 per sample. We removed ASVs that were annotated as chloroplast and mitochondria using a pre-trained taxonomy classifier against the Greengenes (v13_8) 99% Operational Taxonomic Units (OTUs) reference database [39] and further removed rare ASVs with abundance less than 0.0005% of the total number of sequences.

### 3.2. Gut Microbiome Alpha Diversity Analysis

We used two measures Phylogenetic Diversity Index and Species Richness for alpha diversity. *p*-values calculated by a Kruskal–Wallis test for phylogenetic diversity index and species richness for all pairs of groups are summarized in Appendix A. First, OC0 (Old-Control-0 month) and OS0 (Old-SFN-0 month) on the other hand, and YC0 (Young-Control-0 month) and YS0 (Young-SFN-0 month) on the other were expected to show similar alpha diversity, since their fecal samples for OS0 and YS0 were collected before SFN diet was administered. OC0 and OS0, YC0 and YS0 showed a visible difference in both the Phylogenetic Diversity Index and Species Richness (Figure 1, Appendix A). Second, **OS2** (Old-SFN-2 month) was the group with the largest variance of alpha diversities. While the diversity and richness of old controls, young controls and young SFN treated mice decreased considerably after two months, the group **OS2** rather increased both in diversity and richness. Interestingly, the gut microbiome in young mice on SFN diet showed a similar pattern over time to old control mice, which might indicate that SFN diet does not alter the gut microbiome in young animals. Significant differences in alpha diversity were observed between OC2 (Old-Control-2 month) and YC2 (Young-Control-2 month) (Appendix A), but not between **OS2** and YS2 (Young-SFN-2 month). Alpha diversity analysis implies that restoration in alpha diversity of the gut microbiome in old mice with SFN diet from 0 to 2 months may be due to SFN, which led to **OS2** being more similar to YS2 than OC2 in alpha diversity. We had expected to find little difference between OC0 and OS0 and between YC0 and YS0, and indeed there were no significant differences found (Appendix A).

### 3.3. Gut Microbiome Beta Diversity Analysis

We explored four different metrics (weighted-UniFrac, unweighted-UniFrac, Bray–Curtis, and Jaccard) to get a broader view in comparing community structures. Similar to the alpha diversity analysis, we expected small beta diversity between OC0 and OS0, and between YC0 and YS0. Significant differences were observed in unweighted-UniFrac but not in weighted-UniFrac, which might indicate that differences between OC0 and OS0, and between YC0 and YS0 are generally driven by rare organisms. Interestingly, weighted-UniFrac showed different patterns in beta diversity from other distances (unweighted-UniFrac, Jaccard, Bray–Curtis), which indicates that SFN diet changed the microbial community structure by increasing the abundance of specific taxa. Figure 2 shows principal coordinate analysis (PCoA) based on beta diversity measures. The weighted-Unifrac distance accounted for the variance explained the most with the first two principal components (50.9 + 12.8 = 63.7% of the variance) whereas unweighted-UniFrac accounted for 40.9%, Bray–Curtis for 37.9% and Jaccard for 26.8% of the variance. Figure 2 shows that the group **OS2** is clearly separated from the other old groups (OC0, OC2, OS0) and is closer to the groups YC2 and YS2 with weighted-UniFrac, Bray–Curtis, Jaccard measures. SFN-induced beta diversity separation of old group was not obvious with unweighted-UniFrac distance (Figure 2D). Nonetheless, differences between **OS2** and YC2 with all diversity measures considered, and the difference between **OS2** and YS2 with weighted UniFrac were statistically significant (Appendix A). Overall, **OS2** is more similar to YS2 and YC2 than OC2 in beta diversity, which implies that the SFN diet alters and may restore the young gut microbiome in old mice.

### 3.4. Gut microbiome Taxonomic Profiling

To further uncover microbial composition characteristics in different groups, we analyzed ASVs assigned for phylum, class and genus in Figure 3.

At the phylum level (Figure 3A), Firmicutes, Bacteroides and Verrucomicrobia were the most predominant phylum groups. At the class level (Figure 3B), Clostridia (belonging to Firmicutes at the phylum level), Erysipelotrichia (Firmicutes), Verrucomicrobiae (Verrucomicrobia), Bacteroidia (Bacteroidetes) and Bacilli (Firmicutes) were the most predominant classes. At the genus level (Figure 3C), *Akkermansia* (belonging to Verrucomicrobiae at the class level), *Allobaculum* (Erysipelotrichi), *Bacteroides* (Bacteroidia), *Lactobacillus* (Bacilli), *Odoribacter* (Bacteroidia), *Oscillospira* (Clostridia), *Parabacteroides* (Bacteroidetes), *Prevotella* (Bacteroidetes), *Ruminococcus* (Clostridia) and *Turicibacter* (Erysipelotrichi) were predominant genera. The abundances of these microbial taxa changed significantly across groups. At the phylum level, SFN diet increased Firmicutes and decreased Bacteroidetes in old mice. This effect was not noticeable in young mice as compared to old mice. Within the phylum Firmicutes, two classes, Erysipelotrichi and Clostrida, showed the opposite trend in the SFN diet group. Erysipelotrichi occupied a large portion of the gut microbiota in stool sample equally in **OS2**, YC2 and YS2, whereas Clostrida decreased in these groups. Bacteroidia belonging to the phylum Bacteroidetes accounted for a smaller proportion of gut microbiota in **OS2** compared to all other groups, which might be due to SFN administration in old mice. We observed that the YC2 and YS2 groups contained many more sequences annotated at the genus, which might be a characteristic of the gut microbiome in young mice. Interestingly, **OS2** showed a similar number of sequences annotated at the genus to young mice, unlike OC2. We witnessed a marked increase of Erysipelotrichi and *Allobaculum* in young mice over two months (from 0 month to 2 month), both in the SFN diet and control groups. Interestingly, very similar increases of Erysipelotrichi and *Allobaculum* were observed in old mice with SFN diet, but not in old mice with control diet (Figure 3B,C). Therefore, SFN diet alters the old gut microbiome and might play a major role in ensuring that old gut microbiome follows one characteristic of the young gut microbiome by restoring the abundance of *Allobaculum* in the old mice. *Akkermansia* decreased and *Oscillospira* increased in abundance in old and young mice due to the SFN diet. It is noted that young groups at the baseline (YC0 and YS0) somehow showed significant differences in abundance of *Prevotella* and *Bacteroides* (genera of the phylum Bacteroidetes), which might be anticorrelated [40,41]. The genus *Prevotella* is a large genus with high species diversity that contains beneficial bacteria that degrade dietary fiber into short chain fatty acids (SCFAs), although the species *Prevotella copri* has been suggested to be linked to inflammation in HIV [42]. A high *Prevotella*-to-*Bacteroides* ratio is associated with a loss of body weight and body fat [43]. This evidence might support the contention that there is no unique optimal young gut microbiota composition. Based on all the evidence shown above, we conclude that SFN diet restored the young gut microbiome in old mice by increasing the abundance of specific taxa. This recovery of the microbiome may contribute to reversing functional and physiological effects of aging.

### 3.5. Linear Discriminant Effect Size Analysis (LEfSe) of Gut Microbiota

To identify distinctive features between groups, a linear discriminant analysis (LDA) effect size (LEfSe) analysis [33] was performed (Figure 4). LEfSe analysis showed that the genus *Allobaculum*, the order Erysipelotrichales, the class Erysipelotrichi and the phylum Firmicutes were significantly enriched in **OS2** compared to OC2. In addition, LEfSe analysis also showed that the genus *Adlercreutzia*, the order ***Coriobacteriales***, the class Coriobacteriia, and the phylum Actinobacteria were significantly higher in **OS2** compared to OC2. However, the genus *Adlercreutzia* is from the class Actinobacteria and its abundance is negligible compared to other classes and genera in Figure 3B and Figure 4B. None of the significantly enriched features in **OS2** compared to OC2 were identified as enriched features in YC2 and YS2 compared to **OS2** (Appendix A). On the other hand, LEfSe analysis identified the family S24-7, the order Bacteroidales, the class Bacteroidia, and the phylum Bacteroidetes as enriched in OC2 compared to **OS2**.

### 3.6. Potential Functional Annotations of Gut Microbiota in Two Age Groups with and without SFN Diet

We used the PICRUSt2 software [30] for stringent prediction of metagenomes and functional metabolic pathways from the 16S survey data, yielding output as MetaCyc [32] pathway abundances. PCoA analysis was executed with predicted functional profiles by KEGG Orthology [31] (Figure 5A) and metabolic pathways based on the MetaCyc database (Figure 5B). The analyses included only the microbiome data collected at 2 months after initiation of diet treatment. The first two principal components of PCoA explain 43% of variance with the KEGG Orthology profile and 51% of variance with the metabolic pathway profile. Interestingly, both profiles showed an orthogonal relationship between **OS2** and OC2 (Figure 5). In particular, the MetaCyc pathway profile showed that three groups (**OS2**, YC2 and YS2) were horizontally positioned along the first principal component whereas OC2 was positioned along the second principal component (Figure 5B). Figure 5C represents the LEfSe result which displays the effect size of each differentially abundant MetaCyc metabolic pathway between OC2 and **OS2** with LDA score cutoff of 3. Comparing OC2 to **OS2**, we identified 43 significant metabolic pathways among a total of 323. Prominent observations are as follows: amino acid metabolism related pathways, l-lysine biosynthesis II, superpathway of l-phenylalanine biosynthesis, superpathway of l-tyrosine biosynthesis, superpathway of l-lysine l-threonine and l-methionine biosynthesis I, superpathway of l-methionine biosynthesis (transsulfuration), superpathway of *S*-adenosyl-l-methionine biosynthesis were significantly enriched in **OS2** compared to OC2. Two central metabolism-related pathways, incomplete reductive tricarboxylic acid (TCA) cycle and pyruvate fermentation to propanoate I were enriched in OC2 compared to **OS2**.

### 3.7. Gut Metabolome of Two AGE Groups with and without SFN Diet

To determine the metabolic impact of SFN diet in young and old mice, we visualized the overall metabolomic profile for the four comparative study groups (YC2, YS2, OC2 and **OS2**) using partial least square discriminant analysis (Figure 6A). We observed that the old mice on the control diet were well separated from young mice either on control or SFN diet. Interestingly, old mice fed with SFN diet clustered with the young mice and were well separated from old mice (of the same age) on control diet, suggesting that SFN alleviates aging associated metabolic alterations, at least in part. Next, we asked which metabolites in old mice were restored to the levels observed in the young mice cohort. As seen in the heat map (Figure 6B), the levels of metabolites including norepinephrine sulfate, diethyloxipropinate, taurocyamine and 3-hydroxyisovaleryl carnitine were depleted in old mice but restored when the mice were placed on SFN diet.

### 3.8. Microbiome and Metabolome Data Integration Analysis Reveals Microbiome-Dependent Metabolic Changes

We explored two orthogonal approaches, MIMOSA [34] and MelonnPan [35], which are fundamentally different. MIMOSA uses a metabolic model framework that integrates metabolic potential from bacterial genomes and metabolome composition into a unified analysis. Community-wide metabolic potential (CMP) scores are predicted based on the microbial metabolic genes for each metabolite and sample, and then are compared to the actual metabolome data obtained experimentally. MelonnPan calculates the correlation between the microbiome and metabolome composition to identify bacterial populations that might be responsible for metabolite patterns using an elastic regularization technique. Our focus was to identify well-predicted metabolites by changes in the microbial community due to the SFN diet in old mice, which resulted in 565 metabolites among 4158 putative metabolites in negative ion mode and 192 metabolites among 4197 putative metabolites in positive ion mode. First, to perform MIMOSA, we mapped the corresponding metabolite names to KEGG identifiers by mapping the compound IDs to the Human Metabolome Database (HMDB), which includes cross-references to KEGG compound identifiers. Only 10 metabolites out of 565 significant metabolites in negative ion mode and 6 metabolites out of 192 significant metabolites in positive ion mode were identified with unique KEGG compounds. With log-transformed metabolite values, MIMOSA identified only one metabolite, Phosphatidylcholine. Appendix A represents the relationship between the CMP score estimated by a rank-based method and the experimentally measured metabolite value for each sample. Of the metabolite values, 23% (model R-squared of 0.233) was explained by taxa that can contribute to phosphatidylcholines. For phosphatidylcholine, MIMOSA identified 29 contributing taxa, among which 6 taxa were selected based on an abundance cutoff of 10, sample frequency of 20% and contribution of 5% to model R-squared (Appendix A). MIMOSA identified the genus Parabacteroides as the major contributing taxon for phosphatidylcholine, followed by the family S24-7.

To run MelonnPan, we limited our analysis to OTUs which are prevalent in at least 20% of the samples, with relative abundance larger than the mean of abundances of OTUs (=4.209726) and with a variance larger than the mean of variances of OTUs (=63.0079). This resulted in the analysis of 186 OTUs out of a total of 929 OTUs. We normalized the abundance of the selected OTUs and log-transformed metabolite values into relative abundance. MelonnPan fitted a per-metabolite elastic net model to the normalized data and determined the tuning parameters (the elastic net mixing parameter and sparsity parameter) in the elastic net model using 10-fold cross validation, which led to an optimal subset of OTUs whose abundances predict a given metabolite. Metabolites with a Spearman correlation coefficient between experimentally measured metabolite values and predicted metabolite values across samples >0.3 were defined as well-predicted by MelonnPan. For each well-predicted metabolite, we kept only OTUs whose coefficients in the fitted elastic net model were ≥0.0002. Phosphatidylcholine by MIMOSA was also predicted by MelonnPan (data not shown because coefficients in the fitted elastic net model <0.0002). Among well-predicted metabolites, 61 metabolites (listed in Appendix A) had coefficients of organisms larger than 0.0002 in the fitted elastic net models. We kept only organisms which are annotated at the genus level where the same genus occurred more than once for a given metabolite, then we kept the one with the largest absolute coefficient. Figure 7 represents coefficients of organisms for well-predicted metabolites which have no known involvement in drug and plant metabolism. Those metabolites in Figure 7 that showed an increase in concentration by the SFN diet in old mice are shown in Appendix A. Qualitative changes were seen for taurocholic acid 3-sulfate, phaseolic acid, and 2-deoxyguanosine 5-diphosphate. Differential abundance analysis between YC2 and YS2 identified 436 metabolite biomarkers (data not shown). We confirmed that except taurocyamine, other metabolite biomarkers identified comparing OC2 and **OS2** groups (Figure 7) have no similarity with biomarkers identified comparing YC2 and YS2 groups. This implies that the SFN diet-induced metabolite biomarkers in Figure 7 are associated with changes in the microbiome and are specific to old mice. Lastly, the genus *Oscillospira* was the most associated with well-predicted metabolites. The genera that were also majorly associated with well-predicted metabolites were *Ruminococcus gnavus*, *Allobaculum*, and *Akkermansia muciniphila*.

## 4. Discussion

To our knowledge, this study is the first attempt to investigate the impact of SFN on structural and functional changes in the gut microbiota and metabolome in an animal model of aging.

### 4.1. Age-Dependent Microbial Signatures of the Mouse Gut Microbiome

First, in comparing OC2 to YC2, we determined that the gut microbiome of old mice was composed of more bacteria which were not annotated at the genus level, showing much lower taxonomic resolution compared to the microbiota in young mice. Gut microbial alpha diversity, a holistic estimator generally decreases when people age, likely due to changes in physiology, diet, medication, and lifestyles. Decreased diversity is considered an indicator of an unhealthy microbiome and has been linked to different chronic conditions such as obesity and type 2 diabetes [44]. A decline in gut microbiota diversity was also observed in our dataset (Figure 1). As in the human gut microbiome, Firmicutes and Bacteroidetes were the predominating bacterial phyla in our data. We observed a decrease of the ratio of Firmicutes to Bacteroides with age (Figure 3A), which is also seen in aging humans [45] and is known to be associated with several conditions such as weight gain, obesity, insulin resistance, gut permeability, inflammatory bowel disease, and depression [46,47]. At the class level, gram-negative Bacteroidia (Bacteroidetes phylum) and Gram-positive Clostridia, Bacilli, and Erysipelotrichi (all from the Firmicutes phylum) were dominant classes and altered in old mice. In particular, a decreased abundance of Erysipelotrichi in the old gut microbiome was the most visible. At the genus level, beneficial bacteria *Allobaculum*, *Lactobacillus*, *Odoribacter*, and *Ruminococcus* decreased in the old mouse gut microbiome. A study of calorie restriction [48] found that *Allobaculum* and *Lactobacillus* made a significant contribution to the decrease of Firmicutes during aging. At the genus level, we found SCFAs-producing taxa, *Allobaculum*, *Lactobacillus*, *Odoribacter*, and *Ruminococcus* enriched in YC2 compared to OC2. Among them, *Allobaculum* (from the Erysipelotrichi class) is a SCFA-producing genus in the gut which has recently been reported to be an important functional phylotype [49,50] and, especially, protects intestinal barrier function by producing SCFAs. *Ruminococcus* is related to polysaccharide fermentation into SCFAs and bile acid dihydroxylation. The members of the genus *Allobaculum* are already known to be inversely correlated with dietary-induced inflammation markers [51]. *Odoribacter* is a butyrate producer that belongs to the phylum Bacteroidetes [52,53,54,55]. The decrease of these SCFA producers in old-control mice might suggest an increase in the possibility of inflammaging, i.e., increased chronic, low-grade inflammation [56]. The beneficial microbe, *Akkermansia* (Phylum Verrucomicrobia) produces both propionate and acetate [57,58] and is inversely correlated with several disease states. However, at the baseline, *Akkermansia* was somehow enriched in old mice with control diet compared to other groups with SFN diet. The old group with control diet maintained its Akkermansia abundance over time whereas the abundance of Akkermansia decreased in the young group with SFN diet (Figure 3C). Certain members of this genus are mucin degraders (for example, Akkermansia muciniphila) which could exacerbate infection. It would be worthwhile to investigate the effect of SFN diet on the gut microbiome at the strain level.

### 4.2. SFN Diet-Dependent Microbial Signatures in the Mouse Gut Microbiome

Gut microbiota have emerged as an attractive therapeutic target to promote healthy aging and anti-aging effects, which may result in improved quality of life [59]. The gut microbiota responds to dietary interventions very quickly, and short-term changes in the diet can alter the overall structure of the gut microbiota. We have recently shown that an SFN-containing diet reduces the effects of aging on cardiac and skeletal muscle function (unpublished data). In this study, we investigated the impact of SFN diet on the mouse gut microbiota, which may contribute to these anti-aging effects. SFN diet increased the diversity of the gut microbiota in old mice (Figure 1). Furthermore, SFN diet encouragingly reshaped the gut microbial community structure of old mice to where they approach the one of young mice for all four beta diversity measures (Figure 2). Together, these results demonstrate the possibility of anti-aging effects of SFN on the gut microbiota.

At the phylum level, the SFN diet enriched Firmicutes, depleting Bacteroides in old mice, which are known to be associated with a healthy gut microbiome. The ratio of Firmicutes to Bacteroidetes in the human gut microbiota changes with age, undergoing a significant increase from birth to adults and a significant decrease from adults to elderly, to where there is no significant difference between infants and elderly [60]. In our study, the ratio of Firmicutes to Bacteroidetes in young mice fed SFN was similar to the one in young control mice. In old mice, on the other hand, SFN diet increased the ratio of Firmicutes to Bacteroidetes. Our study pointed to the class Erysipelotrichi and the genus *Allobaculum* as major contributors to the SFN-induced enrichment of the phylum Firmicutes in the old gut microbiota. Increased numbers of Erysipelotrichi are associated with a phenotype of impenetrable mucus layer in the mouse [61]. Moreover, exercise increases butyrate-producing Erysipelotrichaceae along with several taxa in animals and adults regardless of diet [62]. *Allobaculum*, one of the most prevalent genera in young gut microbiota (Figure 3C), showed a significant change in abundance with SFN administration in the old mice (Figure 4). *Allobaculum* is a genus of Gram-positive, non-spore-forming bacteria, strictly anaerobic and non-motile with tryptophan-catabolizing functions and suggested to be the most active glucose user [63]. *Allobaculum* has been suggested to be beneficial for host physiology, and its increase was associated with low-fat feeding compared with high-fat diet feeding in a mouse model [64]. *Allobaculum* was depleted in mice with age-related mitochondrial dysfunction [65]. The end products of *Allobaculum* fermentation are SCFAs, specifically butyrate which has been shown to have epigenetic consequences as a histone deacetylase inhibitor, possess anti-inflammatory and anti-carcinogenic properties [66]. Therefore, increases in *Allobaculum* due to the SFN diet may contribute to an increased intestinal barrier function and reduced inflammation in old mice. *Oscillospira*, one of the most abundant genera, is an under-studied anaerobic bacterial genus from Clostridial cluster IV in the Firmicutes phylum. *Oscillospira* showed an increased abundance in **OS2** compared to OC0 mice. A study of *Oscillospira* metabolism [67] suggested that *Oscillospira* species are butyrate producers and found that their abundance decreased in inflammation-related diseases. Given the strong evidence linking *Oscillospira* to human leanness or body mass index in both children and adults [68,69,70,71], further studies are necessary to ascertain the clinical significance of *Oscillospira* in human health and aging. However, we note that although we observed overall changes in SCFA profiles, the changes were not statistically significant between OS2 and OC2.

### 4.3. SFN Diet-Dependent Microbial Functional Signatures in the Mouse Gut Microbiome

PCoA analysis based on Euclidian distance demonstrated that the old group fed the SFN diet showed metabolic pathway profiles that were more similar to the ones of young animals than the mice in the old control group (Figure 5). Therefore, the SFN diet could be reasonably presumed to shape the metabolic capability of the aging gut microbiome toward a younger gut microbiome. Noticeably, the TCA cycle, one of the essential functions of mitochondria [72] and a central hub for energy metabolism and macromolecule synthesis, and pyruvate fermentation to propanoate I were significantly enriched in the old control group. A comparison study of the human gut microbiota between centenarian, elderly and young individuals, based on the premise that centenarians are a model of healthy aging, identified these two metabolic pathways enriched in the centenarian group compared to the young and elderly groups [73]. Since pyruvate is the major precursor for the synthesis of three major SCFAs, acetate, propionate, and butyrate that are related to a strong gut epithelium, this study interpreted the enrichment of the TCA cycle and pyruvate fermentation to propanoate I as a signature of longevity and healthy microbiome. On the other hand, increased metabolism of SCFAs such as pyruvate, butanoate and propanoate in obese rodent models may provide an extra energy source and induce insulin resistance [74]. In the current study, metabolic pathways of amino acids such as l-lysine, l-tyrosine, l-methionine, l-threonine, and l-phenylalanine were significantly enriched in the old group on SFN diet compared to the old control group. Among those amino acids, lysine, tyrosine, methionine, and phenylalanine are suggested to play an important role in the regulation of aging [75]. Interestingly, l-lysine is an essential ketogenic amino acid and critical building block for proteins. An increase in l-lysine in the human gut was also observed in young compared to centenarian and elderly groups [73]. Our metabolic pathway analysis was done with inferred functional profiles. There are questions that still remain unanswered, and may need further study. For example, does SFN diet activate all genes in TCA cycles or l-lysine biosynthesis II, or is there a selectivity for particular ones?

### 4.4. SFN Diet-Dependent Microbiome-Dependent Metabolites

To identify microbe-associated metabolites, we applied metabolic model-based (MIMOSA) and multiple regression with penalty term-based (MelonnPan) methods for our paired data. Both methods identified phosphatidylcholine (C00157) as a metabolite associated with microbiome structure where phosphatidylcholine showed an increased fecal concentration in the old group fed the SFN diet compared to the old control group. Phosphatidylcholine is one of the major phospholipids comprising the cellular membrane and is known to have several health-promoting properties. Phosphatidylcholine showed a lifespan-extending effect under oxidative stress (one of the major causal factors of aging) and delayed age-related decline of motility in a worm model [76]. MIMOSA identified *Parabacteroides* and several members from the family S24-7 as major contributors to phosphatidylcholine. Note that Parabacteroides did not show a significant differential abundance between the SFN-treated and control old mice in the current study. Among 61 differentially abundant metabolites between old SFN-treated and old control mice as identified by MelonnPan (in Appendix A), several are not related to drug or plant metabolism. LysoPE(18:4(6z,9z,12z,15z/0:0) and phaseolic acid showed an increased concentration with SFN treatment in old mice and are in positive association with *Allobaculum*, *Oscillospira*, *Ruminococcus/Ruminoccus gnavus*, where *Allobaculum* is one of the SFN diet-induced anti-aging microbial signatures. Phaseolic acid is a hydroxycinnamic acid. Hydroxycinnamic acids and their derivatives have been reported to possess properties of antioxidant, anti-inflammatory, antimicrobial, and anti-tyrosinase activities [77] which might suggest that phaseolic acid can be exploited as an anti-aging agent. Glycinamide ribonucleotide is an intermediate in de novo purine biosynthesis [78]. It showed an increased concentration in SFN-treated old mice and positive associations with *Adlercreutzia*, *Allobaculum*, and *Oscillospira*, where *Adlercreutzia* and *Allobaculum* were both SFN diet-induced anti-aging microbial signatures. Lastly, Glaucarubin concentration increased with SFN treatment in old mice, and this metabolite was positively associated with *Allobaculum*. Glaucarubin is found in fats and oils. Zarse et al. reported that Glaucarubin may promote metabolic health and lifespan in mammals and possibly humans [79]. Many studies reported protective effects of SFN against brain diseases [80]. Among the SFN diet-induced metabolite signatures, taurocyamine is known as an endogenous alkaline shifter that effectively reduces the extent of brain intracellular lactic acidosis brought about by anoxic insult [81], and is an inhibitor of taurine transport and a glycine receptor antagonist in the brain [82]. Taurocyamine is positively associated with *Allobaculum* and *Ruminococcus*. MelonnPan does not use prior information of genomic metabolic capability and solely depends on abundance of species/functions and metabolite concentration. In the analysis of integrated metabolome/microbiome data, it might be a good strategy to use MelonnPan first to identify a subset of metabolites within untargeted metabolomics data, followed by the use of targeted metabolomics data of those metabolites identified by MelonnPan to investigate the relationship between the microbiome and metabolome using MIMOSA.

## 5. Conclusions

The immune system is likely influenced by the gut microbiota, and their interaction plausibly contributes to the process of inflammation. Ongoing studies suggest that diet has an effect on both the gut microbiota and systemic inflammation, with an impact on functional status of older adults. Manipulating the intestinal microbiota may be beneficial for maintaining health and treating disease and potentially mitigating aging related effects on systemic metabolism. SFN is known to have a wide range of biological effects including anticancer, anti-inflammatory, antioxidant, and anti-aging [11]. Here, we used a mouse model of aging that allowed us to control for confounding factors from human data and investigated the impact of diet supplemented with SFN on the structure and function of the gut microbiota and metabolome. We observed structural and functional changes in the microbiome that correlate with age. We demonstrated that SFN diet restored diversities of young microbiota and metabolome in old mice. In particular, SFN diet enriched bacteria associated with improved intestinal barrier function and anti-inflammatory pathways. Moreover, inferred metagenome-based data analyses revealed that the SFN diet decreased abundance of metabolites of the TCA cycle and increased amino acid metabolism related pathways in old mice. Integrated microbiome and metabolome analysis revealed putative metabolite biomarkers of SFN-induced in old mice that could be modulated by the microbiome. Probiotics have exerted very modest effects on the microbiome in the mice [83,84] relative to that in humans, and even in humans the effectiveness of probiotics is controversial [85]. Consequently, the beneficial modification of the microbiome reported by SFN here should be investigated in humans as an alternative to probiotic use. Our observations support a novel finding that SFN diet exerts its anti-aging effect by influencing the composition and function of the gut microbiota. Many observations we made in our study are consistent with earlier studies of the human gut microbiome and aging, suggesting there may be some parallel shifts that occur in aging human and mouse populations. However, translational benefits of SFN in the human gut microbiome remain to be demonstrated. In addition, further studies are needed to investigate potential gender differences in the effects of SFN diet on shaping the gut microbiota. In conclusion, according to our knowledge, our study represents the first effort to investigate the impact of SFN on the gut microbiome and metabolome during aging, providing a new prospective for potential targets for microbiota-targeted intervention.

## Figures and Tables

**Figure 1 microorganisms-08-01500-f001:**
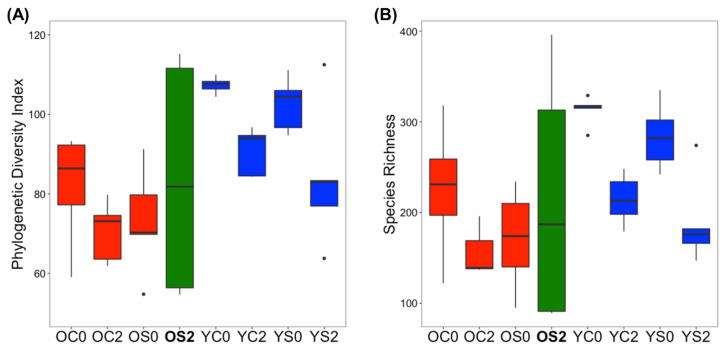
Alpha diversity in the different experimental groups. (**A**) phylogenetic diversity index, and (**B**) species richness. The boxes denote interquartile ranges (IQR) with the median as a black line and whiskers extending up to the most extreme points within 1.5 times IQR. Outliers are noted as points. Please note that O means Old and Y stands for Young; C means control diet and S indicates SFN diet; 0 means 0 months (immediately before diet administration) and 2 stands for 2 months after initiation of diet administration. Alpha diversity significance test results are summarized in Appendix A.

**Figure 2 microorganisms-08-01500-f002:**
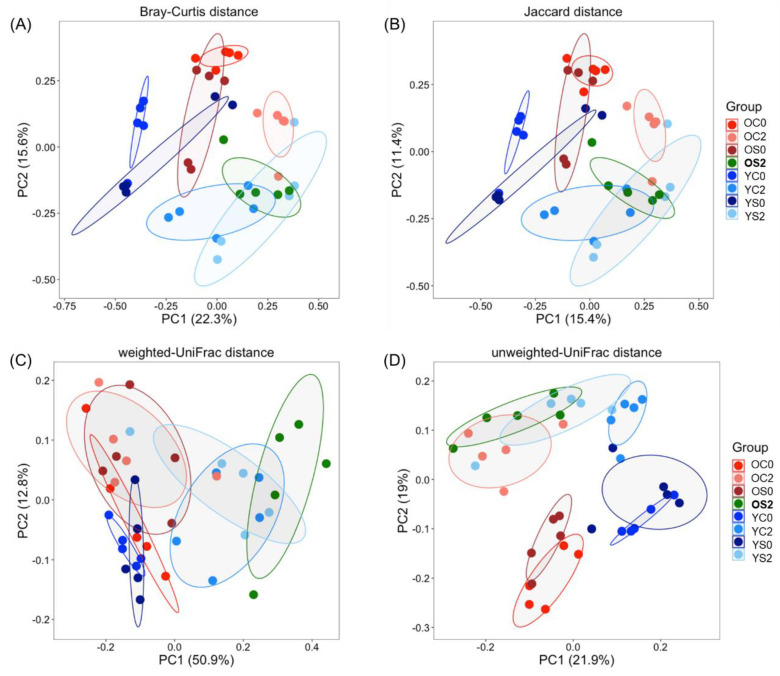
Principal Coordinate Analysis (PCoA) of the gut microbial communities in the different experimental groups. (**A**) Bray–Curtis distance, (**B**) Jaccard distance, (**C**) weighted-UniFrac distance, (**D**) unweighted-UniFrac distance. The individual samples are indicated by symbols according to age (old, young) and color-coded according to the groups. The colored ellipses indicate the 70% confidence interval of each group. The weighted-UniFrac distance accounted for the variance explained the most compared to other diversities. Notations of the groups are the same as in Figure 1.

**Figure 3 microorganisms-08-01500-f003:**
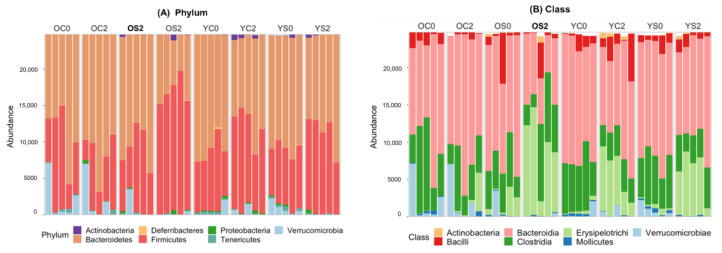
Gut microbial composition (**A**) at the phylum level, (**B**) at the class level, and (**C**) at the genus level. Each bar represents abundance of different kinds of bacteria in gut microbiota of each sample. Group notations are the same as in Figure 1.

**Figure 4 microorganisms-08-01500-f004:**
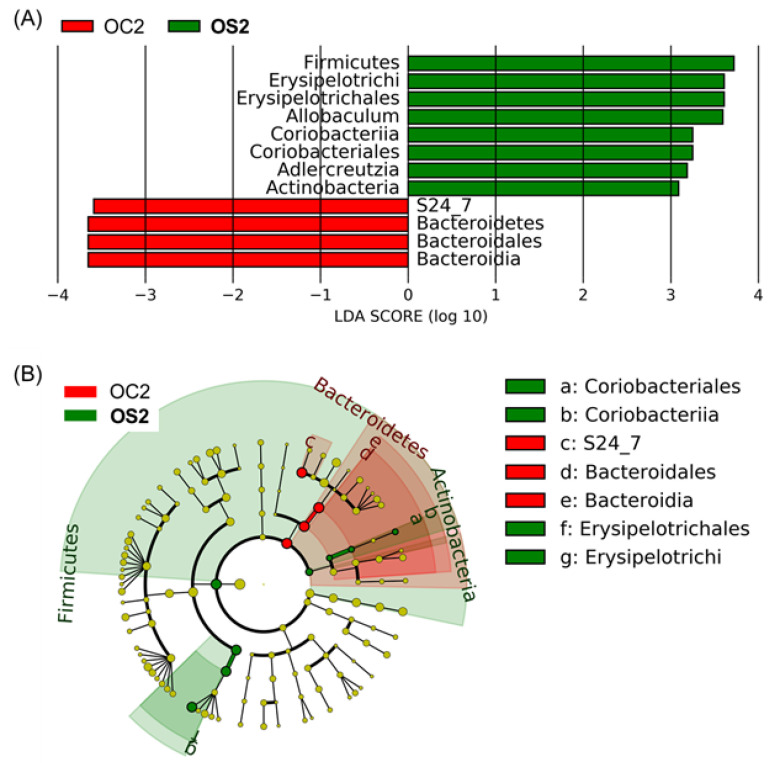
Identification of bacterial biomarkers between SFN-treated old mice (**OS2**) and old control mice of the same age (OC2), using LEfSe analysis. (**A**) The distribution bar chart of LDA values shows the species with linear discriminant analysis (LDA) scores greater than 3 and the species with significantly different abundances in different groups. The length of the histogram represents the impact size of significantly different species. (**B**) The circle radiating from inside to outside represents the classification from the phylum to the genus level. Each small circle at different classification levels represents a sub-classification level. The diameter of the small circle is proportional to the relative abundance. Group notations are the same as in Figure 1. LDA scores < 0 indicate that corresponding features are enriched in OC2.

**Figure 5 microorganisms-08-01500-f005:**
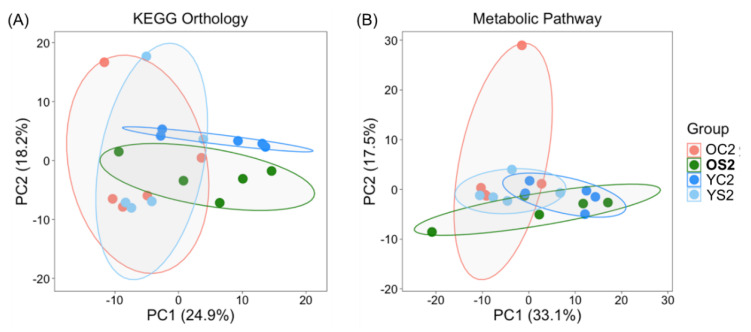
Comparative analyses of the inferred functional profiles of the gut microbiome. (**A**) Principal coordinate analysis (PCoA) of inferred functional profiles by KEGG Orthology in different experimental groups, (**B**) PCoA of inferred metabolic pathway profiles based on the MetaCyc database. The colored ellipses indicate 70% confidence intervals for each group. (**C**) Metabolic pathways with a significant difference between old control mice (OC2) and old mice treated with SFN (**OS2**) (*p* < 0.05 by the Wilcoxon test) using linear discriminant effect size analysis. LDA scores < 0 indicates that corresponding features are enriched in OC2.

**Figure 6 microorganisms-08-01500-f006:**
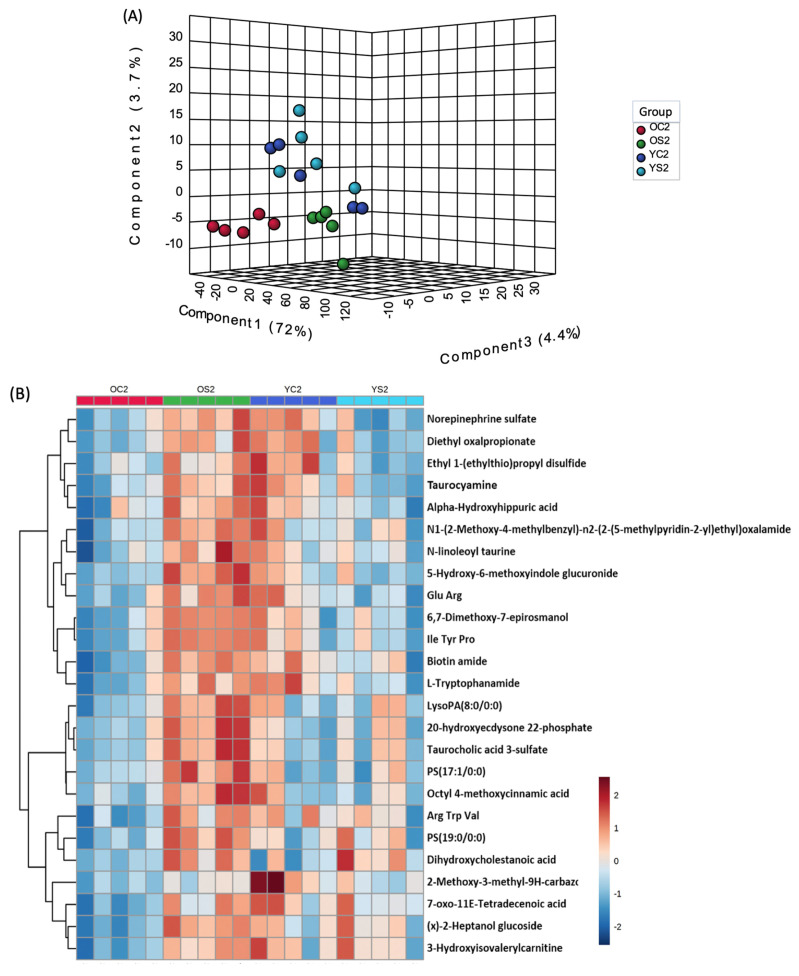
(**A**) A 3D partial least square discriminant analysis (PLS-DA) plot showing group separation based on metabolic profiles. X-axis shows inter-group separation, while Y-axis illustrates intra-group separation. (**B**) Heatmap showing normalized abundance of metabolites across the four study groups (OC2, **OS2**, YC2, YS2).

**Figure 7 microorganisms-08-01500-f007:**
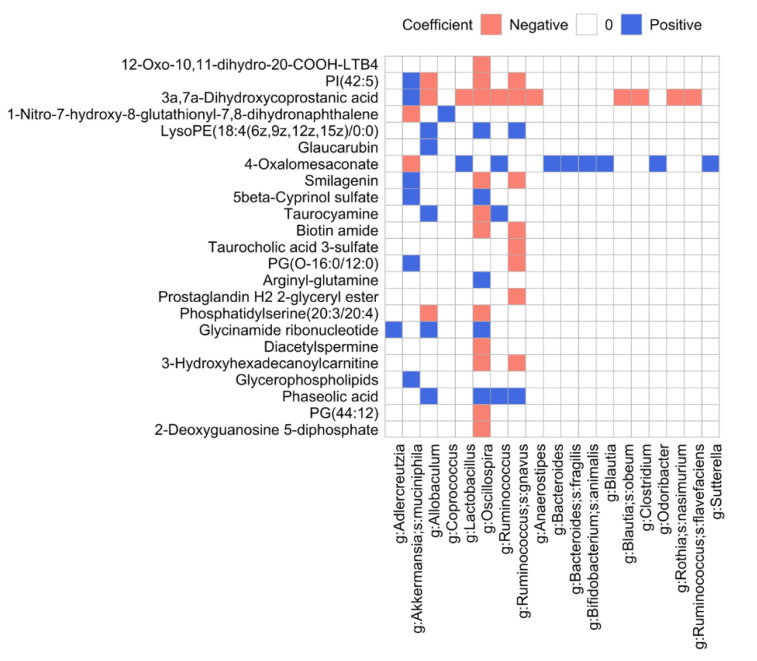
Results of MelonnPan analysis. Comparative analysis of old control and old SFN diet groups using MelonnPan yielded 61 endogenous metabolites showing significant change in abundance, with coefficients of operational taxonomic units in the fitted elastic net models larger than 0.0002. The metabolites shown in the heatmap have no known involvement in drug and plant metabolism.

**Table 1 microorganisms-08-01500-t001:** Experimental design and study groups.

Group	Age	Diet	Time (Month)	Number of Subjects
YC0	Young	Control	0	5
YC2	Young	Control	2	5
YS0	Young	SFN	0	5
YS2	Young	SFN	2	5
OC0	Old	Control	0	5
OC2	Old	Control	2	5
OS0	Old	SFN	0	5
**OS2**	Old	SFN	2	5

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
