# Peer review of "Multi-Omic Analysis Reveals Different Effects of Sulforaphane on the Microbiome and Metabolome in Old Compared to Young Mice"

_microorganisms, 2020, doi:10.3390/microorganisms8101500_

Round 1

Reviewer 1 Report

The article has a scientific value and novelty, is written legibly and all information are presented in a logical way. The introduction describes an actual state of the art in this field. Methodology is good but some explanation should be provided. Mice were fed TD 96,163 diet (Teklad, Madison, WI). Does it contain isoflavone or other flavonoids? Does it contain any natural isothiocyanates from plants? How can it interact with conducted study?

The authors described Prevotella as beneficial microorganisms but Prevotella is often recovered from anaerobic infections of the respiratory tract and gut inflammation (Nature Reviews. Gastroenterology & Hepatology doi:10.1038/nrgastro.2016.4). The Discussion should be extended on better explanation of this phenomenon.

Author Response

Reviewer #1: The article has a scientific value and novelty, is written legibly and all information are presented in a logical way. The introduction describes an actual state of the art in this field. Methodology is good but some explanation should be provided.

We are very grateful for the Reviewer’s appreciation of our findings.

Mice were fed TD 96,163 diet (Teklad, Madison, WI). Does it contain isoflavone or other flavonoids? Does it contain any natural isothiocyanates from plants? How can it interact with conducted study?

The TD 96,163 experimental diet (Teklad, Madison, WI) was essentially antioxidant (TBHQ) free standard AIN-93M diet with sulforaphane. This information was added to the methods section, lines 95-98. The composition of the diet (per 1000g) is: Purified Diet Casein 140g; L-Cystine 1.8g; Corn Starch 465.692g; Maltodextrin 155.0g; Sucrose 100.0g; Soybean Oil 40.0g; Cellulose 50.0g; Mineral Mix, AIN-93M-MX (94049) 35.0g; Vitamin Mix, AIN-93-VX (94047) 10.0g; and Choline Bitartrate 2.5g. For control diet sulforaphane was excluded from the diet mix. The diet does not contain isoflavone or other flavonoids. D,L-sulforaphane was purchased from Toronto Research Chemicals, 20 Martin Ross Avenue, North York, ON, Canada, M3J 2K8 (https://www.trc-canada.com/product-detail/?S699115) that does not contain any other natural isothiocyanates from plants. Therefore, we are confident that any other antioxidants, isoflavone or other flavonoids, and any other natural isothiocyanates from plants were not interacting with the conducted study. We felt this was too much added material to include in the manuscript, and we have only added the information on the source of sulforaphane (lines 95-98) also requested by Reviewer 2. Please let us know if the other information should be added to the Materials and Methods section.

The authors described Prevotella as beneficial microorganisms but Prevotella is often recovered from anaerobic infections of the respiratory tract and gut inflammation (Nature Reviews. Gastroenterology & Hepatology doi:10.1038/nrgastro.2016.4). The Discussion should be extended on better explanation of this phenomenon.

Thank you so much for the reference. We expanded the discussion and added more information on Prevotella in line with the reference (please see in lines 329-332). In these lines, it is noted that young groups at the baseline (YC0 and YS0) somehow showed significant differences in abundance of Prevotella and Bacteroides (genera of the phylum Bacteroidetes) which might be anticorrelated (1, 2). The genus Prevotella is a large genus with high species diversity which contains beneficial bacteria that degrade dietary fiber into short chain fatty acids (SCFAs) although the species, Prevotella copri is suggested to be linked to inflammation in HIV (3).

Reviewer 2 Report

1. In Abstract: “The SFN diet restored the gut microbiome in old mice to mimic that in young mice, enriching bacteria associated with an improved intestinal barrier function and the production of anti-inflammatory compounds.” I cannot find any functional validation on animal. What’s the physiological and biochemistry changes with SFN treatment? Please provide that. There is also lack evidence showing the anti-aging and anti-inflammatory effects of SFN on animals itself. 2. Line 242-244 “Significant differences in alpha diversity were observed between OC2 (Old-Control-2 month) and YC2 (Young-Control-2 month) (Table S1), but not between OS2 and YS2 (Young-SFN-2 month). Alpha diversity analysis of the two different age groups with and without SFN diet implies that SFN may recover the young gut microbiome in old mice.” Alpha diversity only cannot suggest SFN treatment recover to young gut microbiome. It may due to SFN effect, so OS2 is more similar to YS2 than OC2. Similar condition is also seen in beta diversity. 3. Figure 3. The authors claimed that “Erysipelotrichi occupied a large portion of the gut microbiota in stool sample equally in OS2, YC2 and YS2, whereas Clostrida was decreased in these groups.” “Allobaculum was also increased equally in young control and SFN groups at 2 months. Therefore, the genus Allobaculum might be a characteristic of the gut microbiome in young mice, and the SFN diet restored Allobaculum in the old gut microbiome.” However, the high abundance of Erysipelotrichi and Allobaculum is not observed in YC0 and YS0 group. Thus, it is not convincing to say high abundance of Erysipelotrichi and Allobaculum is belong to young microbes. 4. Line 450 “The decrease of these SCFA producers in old-control mice might suggest an increase in the possibility of inflammaging, i.e. increased chronic, low-grade inflammation….” Can the authors provide evidence for SCFA change and anti-inflammatory evidence for SCFA treatment ? 5. line 87. sulforaphane (SFN) was obtained from where and how to prepare and how to treat the animal ? 6. line 455 “Contrary to our expectations, Akkermansia was enriched in old mice (Figure 3C).” It should be explained and discussed. 7. line 531-533 “Among 61 differentially abundant metabolites between old SFN-treated and old control mice as identified by MelonnPan, several are not related to drug or plant metabolism.” I cannot find the 61 differentially abundant metabolites. In which table or figure? 8. What’s the metabolites difference between young and old animals? What’s the metabolites change with SFN treatment. Did SFN treatment recover the metabolites of old animal? Can you illustrated that for authors ?

Author Response

Response to Reviewers:

We thank the Editor for giving us the opportunity to revise the manuscript and the Reviewers for their constructive comments. Based on the comments of the Reviewers, we have now revised the manuscript with additional data as requested. Pointwise responses to the Reviewers’ comments are offered as follows:

Reviewer 2

  1. In Abstract: “The SFN diet restored the gut microbiome in old mice to mimic that in young mice, enriching bacteria associated with an improved intestinal barrier function and the production of anti-inflammatory compounds.” I cannot find any functional validation on animal. What’s the physiological and biochemistry changes with SFN treatment? Please provide that. There is also lack evidence showing the anti-aging and anti-inflammatory effects of SFN on animals itself.

We agree with the reviewer that we have not presented any functional validation and effect of SFN in this manuscript. We have added “known to be” after bacteria in the quote above (line 30 in the revised manuscript).

We have just had a detailed study accepted for publication 1 (title: Sulforaphane prevents age-associated cardiac and muscular dysfunction through Nrf2 signaling in “Aging Cell”). In that paper, we report that SFN may prevent age-related loss of function in the heart and skeletal muscle. Our studies suggest that in the old mice, SFN restored Nrf2 activity, mitochondrial function, cardiac function, exercise capacity, glucose tolerance and activation/differentiation of skeletal muscle satellite cells. Our results suggested that the restoration of Nrf2 activity and endogenous cytoprotective mechanisms by SFN may represent a safe and effective strategy to protect against muscle and heart dysfunction due to aging. This material has been added to the Introduction in lines 58-64.

  1. Line 242-244 “Significant differences in alpha diversity were observed between OC2 (Old-Control-2 month) and YC2 (Young-Control-2 month) (Table S1), but not between OS2 and YS2 (Young-SFN-2 month). Alpha diversity analysis of the two different age groups with and without SFN diet implies that SFN may recover the young gut microbiome in old mice.” Alpha diversity only cannot suggest SFN treatment recover to young gut microbiome. It may due to SFN effect, so OS2 is more similar to YS2 than OC2. Similar condition is also seen in beta diversity.

Thank you so much for this clarification. We revised our statements about observations based on alpha and beta diversity analysis in lines 255-257 and lines 283-285:

Lines 255-257: Alpha diversity analysis implies that restoration in alpha diversity of the gut microbiome in old mice with SFN diet from 0 to 2 months may be due to SFN, which led to OS2 being more similar to YS2 than OC2 in alpha diversity.

Lines 283-285: Overall, OS2 is more similar to YS2 and YC2 than OC2 in beta diversity, which implies that the SFN diet alters and may restore the young gut microbiome in old mice.

  1. Figure 3. The authors claimed that “Erysipelotrichi occupied a large portion of the gut microbiota in stool sample equally in OS2, YC2 and YS2, whereas Clostrida was decreased in these groups.” “Allobaculum was also increased equally in young control and SFN groups at 2 months. Therefore, the genus Allobaculum might be a characteristic of the gut microbiome in young mice, and the SFN diet restored Allobaculum in the old gut microbiome.” However, the high abundance of Erysipelotrichi and Allobaculum is not observed in YC0 and YS0 group. Thus, it is not convincing to say high abundance of Erysipelotrichi and Allobaculum belong to young microbes.

We agree. We have revised our argument on Erysipelotrichi and Allobaculum as follows:

We have witnessed a marked increase of Erysipelotrichi and Allobaculum in young mice over two months (from 0 month to 2 months), both in the SFN diet and control groups. Interestingly, very similar increases of Erysipelotrichi and Allobaculum were observed in old mice with SFN diet, but not in old mice with control diet (Figure 3B and 3C). Therefore, SFN diet alters the old gut microbiome and might play a major role in ensuring that the old gut microbiome follows one characteristic of the young gut microbiome by restoring the abundance of Allobaculum in the old mice, Lines 317-323.

  1. Line 450 “The decrease of these SCFA producers in old-control mice might suggest an increase in the possibility of inflammaging, i.e. increased chronic, low-grade inflammation….” Can the authors provide evidence for SCFA change and anti-inflammatory evidence for SCFA treatment?

We added a reference to support the sentences which the reviewer commented on. This reference presents several associations between a decrease in SCFA producers and inflammaging through an increase in gut leakiness 2, Line 490-492.

  1. line 87. sulforaphane (SFN) was obtained from where and how to prepare and how to treat the animal?

D,L-sulforaphane was purchased from Toronto Research Chemicals (North York, ON, Canada) and added to the mouse diet. This information was added to the methods section in the manuscript (lines 93-96).

  1. line 455 “Contrary to our expectations, Akkermansia was enriched in old mice (Figure 3C).” It should be explained and discussed.

We added more discussion on Akkermansia as follows:

But, at the baseline, Akkermansia was somehow enriched in old mice with control diet compared to other groups with SFN diet. The old group with control diet maintained Akkermansia abundance over time whereas the abundance of Akkermansia was decreased in the young group with SFN diet (Figure 3C). Certain members of this genus (for example, Akkermansia muciniphila) are mucin degraders which could exacerbate infection. It would be worthwhile to investigate the effects of SFN diet on the gut microbiome at the strain level. (lines 491-497)

  1. line 531-533 “Among 61 differentially abundant metabolites between old SFN-treated and old control mice as identified by MelonnPan, several are not related to drug or plant metabolism.” I cannot find the 61 differentially abundant metabolites. In which table or figure?

Thanks for pointing out the missing table. We included Table S3 in Supplementary Materials. It lists 61 metabolites identified by MelonnPan.

  1. What’s the metabolites difference between young and old animals? What’s the metabolites change with SFN treatment. Did SFN treatment recover the metabolites of old animal? Can you illustrate that for authors? 

To respond to this comment, we performed partial least square discriminant analysis (PLS-DA) based on metabolome data of four groups, OC2, OS2, YC2, and YS2. We added PLSDA results as follows in lines 387-404 and added new Figure 6 A & B.

In order to determine the metabolic impact of SFN diet in young and old mice, we visualized the overall metabolomic profile for the four comparative study groups (YC2, YS2, OC2 and OS2) using partial least square discriminant analysis (Figure 6A). We observed that the old mice on control diet were well separated from young mice either on control or SFN diet. Interestingly, old mice fed with SFN diet clustered with the young mice and were well separated from old mice (of the same age) on control diet, suggesting that SFN alleviates aging associated metabolic alterations, at least in part. Next, we asked which metabolites in old mice were restored to the levels observed in the young cohort. As seen in the heat map (Figure 6B), the levels of several metabolites including norepinephrine sulfate, diethyloxipropinate, taurocyamine and 3-hydroxyisovaleryl carnitine were depleted in old mice but restored when the mice were placed on SFN diet.

Furthermore, we added more discussion on SFN-diet induced metabolite biomarkers in old mice as follows:

Those metabolites in Figure 7 that showed an increase in concentration due to the SFN diet in old mice are shown in Figure S3. Qualitative changes were seen for taurocholic acid 3-sulfate, phaseolic acid, and 2-deoxyguanosine 5-diphosphate. Differential abundance analysis between YC2 and YS2 identified 436 metabolite biomarkers (data not shown). We confirmed that except taurocyamine, other metabolite biomarkers identified comparing OC2 and OS2 groups (Figure 7), has no similarity with biomarkers identified comparing YC2 and YS2 groups. This implies that the SFN diet-induced metabolite biomarkers in Figure 7 are associated with changes in the microbiome and are specific to old mice, lines 448-452.

  1. Bose, C.; Alves, I.;  Singh, P.;  Palade, P. T.;  Carvalho, E.;  Børsheim, E.;  Jun, S.-R.;  Cheema, A.;  Boerma, M.;  Awasthi, S.; Singh, S. P., Sulforaphane prevents age-associated cardiac and muscular dysfunction through Nrf2 signaling. Aging Cell 2020, In Press.
  2. Biragyn, A.; Ferrucci, L., Gut dysbiosis: a potential link between increased cancer risk in ageing and inflammaging. The Lancet. Oncology 2018, 19 (6), e295-e304.

Round 2

Reviewer 2 Report

The manuscript is much improved now. I have several minor points only

  1. I cannot find the reference 9 on web. The manuscript is important since it provide the anti-aging effect of sulforaphane. It should be confirmed.
  2. The title is "Multi-omic analysis reveals the anti-aging impact of sulforaphane on the microbiome and metabolome". But there is no any anti-aging effect showing in this manuscript. The metabolome provided here is related to gut microbiome but not from host directly. The title can be revised or authors can provide direct evidence from host rather than just gut microbiome ?
  3. The authors discuss about short-chain fatty acid, what's the real data for short-chain fatty acid with the sulforaphane treatment (not just related microbiome change) ?

Author Response

Response to Reviewers:

We thank the Editor for giving us the opportunity to revise the manuscript and the Reviewers for their constructive comments. Based on the comments of the Reviewers, we have now revised the manuscript with additional data as requested. Pointwise responses to the Reviewers’ comments are offered as follows:

Reviewer 2:

The manuscript is much improved now. I have several minor points only.

  1. I cannot find the reference 9 on web. The manuscript is important since it provide the anti-aging effect of sulforaphane. It should be confirmed.

The reference 9 below was accepted for publication on August 30, 2020 and is expected to be available online soon. Furthermore, we are sending copy of our accepted manuscript and forwarding acceptance letter to Mr. Nelson Liang (Assistant Editor, Microorganisms).

Bose, C.;  Alves, I.;  Singh, P.;  Palade, P. T.;  Carvalho, E.;  Børsheim, E.;  Jun, S.-R.;  Cheema, A.;  Boerma, M.;  Awasthi, S.; Singh, S. P., Sulforaphane prevents age-associated cardiac and muscular dysfunction through Nrf2 signaling. Aging Cell 2020, In Press.

  1. The title is "Multi-omic analysis reveals the anti-aging impact of sulforaphane on the microbiome and metabolome". But there is no any anti-aging effect showing in this manuscript. The metabolome provided here is related to gut microbiome but not from host directly. The title can be revised or authors can provide direct evidence from host rather than just gut microbiome ?

We revised the title as follows:

Multi-omic analysis reveals different effects of sulforaphane on the microbiome and metabolome in old compared to young mice.

Accordingly, we updated the manuscript by revising sentences which contain anti-aging.

  1. The authors discuss about short-chain fatty acid, what's the real data for short-chain fatty acid with the sulforaphane treatment (not just related microbiome change) ?

Although we observed SCFA-producing taxa which are enriched in OS2 compared to OC2, we didn’t identify SCFAs which are significantly enriched in OS2 compared OC2.

To respond to this comment, we added a sentence in section 4.2 in line 537-539:

However, we note that although we observed overall changes in SCFA profiles, the changes were not statistically significant between OS2 and OC2.

Round 3

Reviewer 2 Report

About the SCFA profiles, although the authors found the changes arenot statistically significant between OS2 and OC2, I suggest provide as supplementary data.